# MiR-146a-5p Expression in Peripheral CD14^+^ Monocytes from Patients with Psoriatic Arthritis Induces Osteoclast Activation, Bone Resorption, and Correlates with Clinical Response

**DOI:** 10.3390/jcm8010110

**Published:** 2019-01-17

**Authors:** Shang-Hung Lin, Ji-Chen Ho, Sung-Chou Li, Jia-Feng Chen, Chang-Chun Hsiao, Chih-Hung Lee

**Affiliations:** 1Department of Dermatology, Kaohsiung Chang Gung Memorial Hospital and Chang Gung University College of Medicine, Kaohsiung 833, Taiwan; hong51@cgmh.org.tw (S.-H.L.); jichenho@cgmh.org.tw (J.-C.H.); 2Graduate Institute of Clinical Medical Sciences, College of Medicine, Chang Gung University, Taoyuan 333, Taiwan; 3Division of Basic Medical Sciences, Department of Nursing, Chang Gung University of Science and Technology—Chiayi Campus, Chiayi County 613, Taiwan; 4Department of Dermatology, Chai-Yi Chang Gung Memorial Hospital, Chiayi 613, Taiwan; 5Genomics and Proteomics Core Laboratory, Kaohsiung Chang Gung Memorial Hospital and Chang Gung University College of Medicine, Kaohsiung 833, Taiwan; raymond.pinus@gmail.com; 6Division of Rheumatology, Allergy and Immunology, Department of Internal Medicine, Kaohsiung Chang Gung Memorial Hospital and Chang Gung University College of Medicine, Kaohsiung 833, Taiwan; uporchidjfc@gmail.com; 7Center for Shockwave Medicine and Tissue Engineering, Kaohsiung Chang Gung Memorial Hospital, Kaohsiung 833, Taiwan

**Keywords:** miR-146a-5p, osteoclast, psoriatic arthritis

## Abstract

In psoriatic arthritis (PsA), progressive bone destruction is mediated by monocyte-derived osteoclasts. MicroRNAs (miRNAs) regulate many pathophysiological processes; however, their function in PsA patient monocytes has not been examined. This study aims to address whether specific miRNAs in CD14^+^ monocytes and monocyte-derived osteoclasts cause active osteoclastogenesis in PsA patients. Candidate miRNAs related to monocyte activation (miR-146a-5p, miR-146b-5p and miR-155-5p) were measured in circulatory CD14^+^ monocytes collected from 34 PsA patients, 17 psoriasis without arthritis (PsO) patients, and 34 normal controls (NCs). CD14^+^ monocytes were cultured with media containing TNF-α and RANKL to differentiate into osteoclasts. Osteoclast differentiation and bone resorption were measured by TRAP immunostaining and dentin slice resorption, respectively. The results showed that the miR-146a-5p expression was higher in PsA patient-derived CD14^+^ monocytes compared to PsO and NCs. Activation and bone resorption were selectively enhanced in osteoclasts from PsA patients, but both were abrogated by RNA interference against miR-146a-5p. More importantly, after clinical improvement using biologics, the increased miR-146a-5p expression in CD14^+^ monocytes from PsA patients was selectively abolished, and associated with blood CRP level. Our findings indicate that miR-146a-5p expression in CD14^+^ monocytes derived from PsA patients correlates with clinical efficacy, and induction of osteoclast activation and bone resorption.

## 1. Introduction

Psoriatic arthritis (PsA) is a chronic inflammatory disease that results in enthesitis, bone resorption, and functional disabilities. It has been estimated that approximately 30% of patients with psoriasis, a chronic inflammatory skin disease, simultaneously suffer from PsA. Skin manifestations usually precede the onset of PsA by 10 years [1]. PsA can involve musculoskeletal structures (joints, entheses, synovial sheaths of tendons, and axial skeleton), skin, nails, and eyes [2]. Approximately half of the patients with PsA suffer from erosive and deforming structural damages within the first two years, causing functional deterioration; these patients also have a higher risk of metabolic syndrome and cardiovascular disease [3,4]. However, PsA is often clinically overlooked in patients with psoriasis due to the subtle initial changes. Haroon et al. reported that for PsA patients, diagnostic delays exceeding six months result in poor radiographic and functional outcomes [5]. Thus, prompt diagnosis and early treatment are crucial for structural damage prevention [6]. Classification Criteria for Psoriatic Arthritis (CASPAR) is the most widely used diagnostic criteria for PsA [7]; it is based on the recognition of clinical and imaging features. However, to date, no standardized and reliable assessment method has been developed for early PsA detection, resulting in an inaccurate estimation of disease state and thereby contributing to the progressive nature of PsA [8].

Pathophysiologically, PsA manifests as osteoclast-mediated focal bone erosions. Osteoclasts, the main cells responsible for bone resorption [9], are derived from monocyte/macrophage lineage precursors [10], which behave as an appropriate and readily accessible source [11,12]. Ritchlin et al. reported a marked increase in osteoclast precursors in PsA patients when compared to healthy controls [13]. Raimondo et al. demonstrated that IL-33, osteopontin, IL-17, and TNF-α induced the release of pro-osteoclastogenic factors—such as RANKL—from the skin of PsA patients, which promote monocyte differentiation into osteoclasts [14].

PsA is a heritable disease with distinct clinical features [15]. Frequency variations in specific subtypes of HLA-B*08, B*27, B*38, and B*39 have been linked to sub-phenotypes in PsA patients, including symmetric or asymmetric axial disease, enthesitis, dactylitis, and synovitis [16]. Upregulation of type I interferon (IFN) inducible genes, and those associated with Th17 cells, which are typical during autoimmunity, confer autoimmune disease properties to PsA [17]. 

MicroRNAs (miRNAs) are a class of small noncoding RNAs that function as important epigenetic regulators. miRNAs expression and functional regulation has been reported in several inflammatory and autoimmune diseases, such as in rheumatoid arthritis (RA), multiple sclerosis, systemic lupus erythematosus, psoriasis, and systemic sclerosis [18]. These studies highlight the potential use of miRNAs as disease severity markers for immunological disorders, by focusing on detecting changes in circulating miRNAs. This represents an attractive methodology due to the noninvasive nature of blood collection, and relative ease of high-throughput detection systems for processing and analyzing its miRNA content. Several studies have investigated the circulating miRNA profiles of PsA patients’ peripheral blood mononuclear cells (PBMC). Pelosi A et al. reported decreased miR-126-3p expression in PBMC’s from active PsA patients [19]. Ciancio et al. reported elevated miR-21-5p levels in PBMC’s isolated at early PsA stages, followed by a significant decrease after successful therapy [20]. However, the regulatory mechanisms by which PBMC-derived microRNAs contribute PsA disease pathogenesis remain unclear. 

miRNAs expression has not been profiled in osteoclast precursors—CD14^+^ monocytes. Previous studies have reported on miRNAs—including miR-155-5p, miR-146a-5p, and miR-146b-5p—mediating monocyte differentiation into macrophages in U937 and THP-1 monocyte/macrophage cell lines [21,22]. In fact, more than 40 miRNAs (including miR-21, -29, -31, -124, -133a, -146a, -223, -503, -378, -125a, -148a, -155, and -422a) have been identified as regulators of osteoclast precursors differentiation into mature osteoclasts [23]. Other miRNAs, in conjunction with known transcription factors and epigenetic regulators, have also been reported to regulate osteoblast differentiation; these include miR-155, -30, -503, -146a, and -541 [24]. Furthermore, Mir-146a has been shown to inhibit proliferation and induce apoptosis in osteoblasts through bcl-2 [25]. Another study published in *PNAS* reported upregulated miR-155 levels in CD68^+^ macrophages derived from the synovium of RA patients [26]. We, therefore, decided to study the role of two common miRNAs—miR-155 and miR-146a—during osteoclastic and osteoblastic differentiation in PsA patients, which is characterized by both osteoclastic and osteoblastic activation. We chose miR-146b—similar form of miR-146a—as an internal control. In the present study, we aimed to investigate the role of miRNA expression in circulatory CD14^+^ monocytes during PsA disease progression.

## 2. Materials and Methods

### 2.1. Study Subjects

This study was approved by the Institutional Review Board. It included 34 PsA patients and 17 psoriatic patients without arthritis, who were confirmed by both dermatologists and rheumatologists. All PsA patients fulfilled the CASPAR criteria. Thirty-four age- and gender-matched healthy adults were included to represent the control group (NC). Thorough examination of all subjects in NC confirmed the absence of psoriatic lesions and inflammatory joint pain. The Psoriasis Area and Severity Index (PASI), C-Reactive Protein (CRP), treatment regimens, comorbidities of arthritis, and presence of enthesitis or uveitis were recorded. Peripheral blood was acquired from all participants at baseline and after 28 weeks of standard biological treatment (etanercept, adalimumab, ustekinumab, or secukinumab).

### 2.2. Isolation and Culture of Peripheral Monocytes

Monocytes were isolated directly from PBMCs using CD14^+^ MicroBeads (Miltenyi Biotec, Auburn, CA, USA) according to the manufacturer’s instructions. We have previously confirmed, using flow cytometry, that the purity of the CD14^+^ cells after selection was about 96.4% [27].

### 2.3. Osteoclast Formation

Purified human CD14^+^ monocytes were seeded at a density of 3 × 10^5^ cells/well onto 96-well plates containing α-MEM with FBS (10%, Invitrogen, Waltham, MA, USA) and M-CSF (20 ng/mL; PeproTech, Rocky Hill, NJ, USA) for 3 days. RANKL (100 ng/mL; PeproTech, Rocky Hill, NJ, USA) and TNF-α (100 ng/mL; PeproTech, Rocky Hill, NJ, USA) were added to induce osteoclast differentiation. Osteoclasts were stained with tartrate-resistant acid phosphatase (TRAP) on day 13 using the Acid Phosphate Leukocyte Kit (Sigma, St. Louis, MO, USA), according to the manufacturer’s instructions. TRAP-stained cells containing three or more nuclei were defined as osteoclasts [28]. The number of osteoclasts was counted from four high power field (HPF; 100×) images per well; then, average was calculated.

### 2.4. Bone Resorption Assay

Purified human monocytes were seeded at 5 × 10^4^ cells/well on dentine slices (IDS, Gaithersburg, MD, USA) in 96-well plates containing α-MEM with 10% FBS and M-CSF (20 ng/mL) for 72 h. The M-CSF-treated cells were incubated with RANKL and TNF-α (100 ng/mL each) to induce osteoclast differentiation. On day 13, the total area of resorption pits in dentine slices was measured under a bright field microscope (Leica DM2500, Wetzlar, Germany). The number of resorption pits was measured using ImageJ software (NIH, Bethesda, MD, USA) from four randomly-selected HPFs.

### 2.5. Transient Transfection of miR-146a-5p Inhibitors

Isolated CD14^+^ monocytes were cultured in α-MEM with 10% FBS and M-CSF for 72 h in 96-well plates on dentine slices. Cells were then transfected with 10 nmol hsa-miR-146a-5p hairpin inhibitor or 10 nmol miRNA hairpin inhibitor as a negative control (Dharmacon, Lafayette, CO, USA) using lipofectamine 3000 for 6 h, based on the manufacturer’s instructions (Invitrogen, Carlsbad, CA, USA).

### 2.6. Quantitative Real-Time PCR Analysis for miRNAs

First strand cDNA was synthesized from RNA samples (100 ng per run) using a TaqMan MicroRNA Reverse Transcription kit (Applied Biosystems; Thermo Fisher Scientific, Inc, Carlsbad, CA, USA), according to the manufacturer’s protocol. Expression profiles including miR-146a-5p (Assay ID. 000468), miR-146b-5p (Assay ID. 001097), and miR-155-5p (Assay ID. 002623) were examined using TaqMan microRNA assays (Applied Biosystems; Thermo Fisher Scientific, Inc.). miRNA-specific primer sequences were designed and synthesized based on the miRNA sequences obtained from the miRBase database: hsa-miR-146a-5p, UGAGAACUGAAUUCCAUGGGUU; hsa-miR-146b-5p, UGAGAACUGAAUUCCAUAGGCU; and hsa-miR-155-5p, UUAAUGCUAAUCGUGAUAGGGGU.

Quantitative real-time PCR (qRT-PCR) was performed on an Applied Biosystems QuantStudio 7 Flex system (Applied Biosystems; Thermo Fisher Scientific, Inc, Carlsbad, CA, USA). Target gene expression levels were normalized to U6 (Assay ID. 4427975). The relative quantity (RQ) of miRNA in each sample was determined by the 2^−ΔΔ*C*_t_^ method, where Δ*C*_t_ = (triplicate *C*_t_ average of the gene target miRNA − triplicate *C*_t_ average of the endogenous control (U6)), and the ΔΔ*C*_t_ = (Δ*C*_t_ − mean Δ*C*_t_ of all samples).

### 2.7. Statistical Analysis

Age, sex, treatment regimen, disease duration, PASI, number of tender or swollen joints, miRNA expression level, number of osteoclasts formed, and the resorption area were compared among groups using Chi-square or *t*-tests based on data normality. The Pearson correlation coefficient was used to measure expression of miR-146a-5p, CRP, and PASI correlation. A *p*-value less than 0.05 was considered statistically significant for all tests.

## 3. Results

### 3.1. Subject Demographics

The study cohorts comprised of: 34 PsA patients (Male/Female: 25/9, and average age: 47.8 years old), 17 psoriatic patients without arthritis (Male/Female: 12/5, and average age: 43.2 years old), and 34 normal controls (NCs; male/female: 19/15, and average age: 44.3 years old) (Table 1). The average PASI calculated for psoriatic patients without arthritis and PsA patients were 15.1 and 15.0, respectively. All PsA patients suffered from peripheral arthritis, including 38.2% with axial arthritis, 25.8% with dactylitis, and 41.9% with enthesitis. Among PsA patients, 23.5% received anti-TNF-α therapy, 29.4% received leflunomide treatment, and 82.4% received methotrexate treatment.

### 3.2. Upregulation of miR-146a-5p in CD14^+^ Monocytes from PsA Patients

In the pilot study, monocyte activation-related microRNA (miR-146a-5p, miR-146b-5p and miR-155-5p) expression levels were measured in CD14^+^ monocytes isolated from 10 patients with severe PsA, and 10 age- and sex-matched NCs. The results showed that while the expression levels of miR-146b-5p and miR-155-5p were similar in both groups, miR-146a-5p expression was five-fold higher in PsA patients compared to NCs (Figure 1).

### 3.3. The Expression of miR-146a-5p was Significantly Increased in CD14^+^ Monocytes from PsA Patients Compared to PsO Patients and NCs

To validate the pilot study data regarding the elevated of miR-146a-5p expression, the expression analysis was validated in a larger cohort; 34 PsA patients, 17 psoriatic patients without arthritis, and 34 NCs, using qRT-PCR and U6 as the internal primer control. As expected, miR-146a-5p expression was significantly higher in CD14^+^ monocytes derived from PsA patients compared to psoriatic patient without arthritis or NCs (ORs: 1.94 (95% CI: 1.25–3.00), *p* < 0.001; Figure 2A). The Δ*C*_t_ values of miR-146a-5p in the NC and PsA groups are summarized in Appendix A. To investigate if miR-146a-5p levels could distinguish PsA patients from NCs, we used support vector machine (SVM) learning [29,30] to calculate the discrimination power of miR-146a-5p. We generated a SVM model with an area under the receiver operating curve (auROC) of 0.78 (Figure 2B), indicating that miR-146a-5p expression could distinguish PsA patients from NCs in 78% of the cases.

### 3.4. The Increased Osteoclast Differentiation and Enhanced Bone Resorption Activity in PsA Patients are Abrogated by RNA Interference against miR-146a-5p

We then investigated if osteoclast activation and bone resorption were enhanced in PsA patients, and whether the miR-146a-5p was involved in the process. To this end, we used RNA interference to generate miR-146a-5p knock down cells. For this experimental model, peripheral CD14^+^ monocytes from 10 PsA patients and five NCs were treated with M-CSF for 72 h. PsA patient cells were then divided into three treatment groups: (1) transfected with a control microRNA inhibitor (negative control); (2) transfected with a miR-146a-5p inhibitor; and (3) non-transfected control. After transfection, cultures were treated with TNF-α and RANKL every three days, for nine days, to induce osteoclast formation. On day 13, the number of generated osteoclasts was lower in the NC (2.2 ± 1.6/HPF) and miR-146a inhibitor groups (2.8 ± 0.8/HPF) compared to the non-transfected and negative control groups (8.8 ± 2.2/HPF and 7.9 ± 3.9/HPF, respectively; Figure 3A,C). Similarly, the resorption activity was lower in the NC group (1.7% ± 2.2%). Bone resorption in the group transfected with siRNA against miR-146a-5p (3.6% ± 1.8%) was significantly lower compared to the non-transfected and negative control groups (7.9% ± 2.9% and 7.6% ± 1.5%, respectively; Figure 3B,D). To test whether osteoclast numbers were associated with the expression of miR-146a-5p in PsA patients, we calculated the R Square values of PsA and NC groups, which were 0.4821 (*p* = 0.08) and 0.2425 (*p* = 0.41), respectively. These data indicate a small but significant association between miR-146a-5p expression and the number of osteoclasts in PsA patients (Figure 3E).

### 3.5. miR-146a-5p Expression was Reduced in CD14^+^ Cells of PsA Patients in Clinical Remission

To clinically validate the role of miR-146a-5p during disease progression, we investigated whether miR-146a-5p expression increase in CD14^+^ monocytes of PsA patients would resume after successful treatment. In the 10 PsA patients that met the ACR20 achievement after 28 weeks of biological treatment (etanercept, adalimumab, ustekinumab, or secukinumab; Figure 4A,B), miR-146a-5p expression levels in CD14^+^ monocytes became comparable to those in NCs (*n* = 10). On the other hand, miR-146b-5p and miR-155-5p expression levels in CD14^+^ monocytes did not change, regardless of clinical improvement after successful treatment (Figure 4C).

### 3.6. The Expression of miR-146a-5p in CD14^+^ Monocytes from Patients with PsA correlates with Blood CRP Level, but not the PASI Score nor the Presence of Enthesitis

Our results indicate that miR-146a-5p expression in CD14^+^ monocytes of PsA patients decreased after clinical improvement following successful treatments. We then investigated the association between miR-146a-5p expression in CD14^+^ monocytes and blood CRP levels in 22 PsA patients. The results showed that the *R* Square value was 0.3645 (*p* = 0.0029; Figure 5A), indicating a small but significant association between the two variables. We also tested the association of miR-146a-5p expression with PASI severity in PsA or PsOs patients. The association was analyzed in 34 patients with PsA, and 17 patients with PsO. The result showed that the R Square values in PsA and PsO patients were 0.17 (*p* = 0.08) and 0.07 (*p* = 0.50), respectively (Figure 5B), indicating a no association of miR-146a-5p expression with PASI severity scores in patients with PsA or PsO. We also applied a similar analysis to determine the association between miR-146a-5p expression in PsA patients and enthesitis. The expression of miR-146a-5p was measured in 12 PsA patients with enthesitis, and 22 PsA patients without enthesitis. The result showed that miR-146a-5p expression was similar in patients with or without enthesitis (*p* = 0.35; Figure 5C).

## 4. Discussion

We present the first study investigating the profile of miRNAs in peripheral CD14^+^ monocytes from PsA patients. miR-146a-5p expression was higher in CD14^+^ monocytes derived from PsA patients compared to NCs. Activation and bone resorption were enhanced in osteoclasts from PsA patients, but both were abrogated by RNA interference against miR-146a-5p. The increased expression of miR-146a-5p in PsA was resumed by successful clinical treatment. 

Pathological bone resorption in PsA results from increased osteoclast precursor numbers [13]. Our study demonstrates that increased miR-146a-5p expression in CD14^+^ monocytes from PsA patients contributes to greater osteoclast formation potential, and active resorption activity. Therefore, measuring miR-146a-5p in CD14^+^ monocytes could be a useful indicator of clinical improvement status. Furthermore, miR-146a-5p expression in CD14^+^ monocytes from PsA patients associated with CRP levels.

miRNA-146a expression strongly increased during osteoclast differentiation in TNF-α and RANKL-treated RAW264.7 cells, it is also reported to inhibit both TNF-α and RANKL-induced osteoclast differentiation by targeting TRAF6 and IRAK1 [31,32]. Li et al. reported that miR-146a expression was highly up-regulated in systemic juvenile idiopathic arthritis monocytes through regulating monocyte polarization [33].

In PsA patients, pro-inflammatory cytokines, such as IL-33, osteopontin, IL-17, and TNF-α—through activation of RANKL, an essential cytokine for osteoclast differentiation—play an important role in osteoclast differentiation and activation [34,35,36,37]. In this study, we detected abnormally high miR-146a expression in CD14^+^ monocytes isolated from PsA patients. Taken together, with the fact that miR-146a may negatively regulate osteoclastogenesis, this study indicates that endogenous miR-146a may play an important role in controlling monocyte-derived osteoclast differentiation, activation, and function, in patients with PsA. 

Selectively targeting specific cytokines (TNF-α, IL-17 and IL-12/23) and/or intracellular signaling pathways effectively prevent disease progression by ameliorating cutaneous inflammation of psoriasis and inhibiting osteoclastogenesis.

Pivarcsi et al. reported that the anti-TNF-α therapy in psoriatic patients decreased serum levels of inflammation- and autoimmunity-related miRNAs, including miR-223, miR-142-3p, and miR-106b [38]. In this study, we measured miR-146a-5p expression in CD14^+^ monocytes (precursor of osteoclasts) before and after successful biological therapy. Our results show that the elevated miR-146a-5p expression in CD14^+^ monocytes from PsA patients, became comparable to those of NCs after successful clinical improvement.

In this study, we show that miR-146a-5p expression correlates with CRP levels. CRP, the most common inflammatory and acute phase protein indicator, is commonly used for evaluating disease activity in rheumatoid arthritis, although half the patients have normal CRP levels [39]. miR-146a-5p expression is not associated with the presence of enthesitis or skin PASI score. However, its expression is associated with blood CRP levels and osteoclasts numbers, making it a good clinical severity indicator. 

This study has several limitations. First, the cohort size was small; therefore, additional large-scale studies are required for validation and to identify potential confounders. Second, some patients received immunosuppressants, which may have affected the miRNAs profile in CD14^+^ monocytes, independent of disease state. Again, a larger patient group, with greater clinical homogeneity, may help further validate our results. 

## 5. Conclusions

Elevated expression of miR-146a-5p in CD14^+^ monocytes isolated from PsA patients is associated with enhanced activation potential and resorption activity. Furthermore, elevated miRNA-146a-5p expression in CD14^+^ monocytes from moderate to severe PsA patients was resolved during clinical remission.

## Figures and Tables

**Figure 1 jcm-08-00110-f001:**
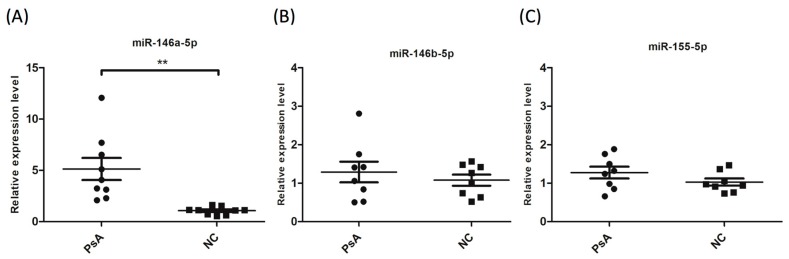
miR146a-5p expression is higher in CD14^+^ monocytes from PsA patients compared to normal controls (NCs). RNA samples were extracted from peripheral CD14^+^ monocytes of PsA patients and NCs. The expression levels of (**A**) miR-146a-5p, (**B**) miR-146b-5p, and (**C**) miR-155-5p were measured in NCs (*n* = 10) and PsA patients (*n* = 10) by qRT-PCR. Patients with PsA showed increased expression of miR-146a-5p in CD14^+^ monocytes. ** *p* < 0.001.

**Figure 2 jcm-08-00110-f002:**
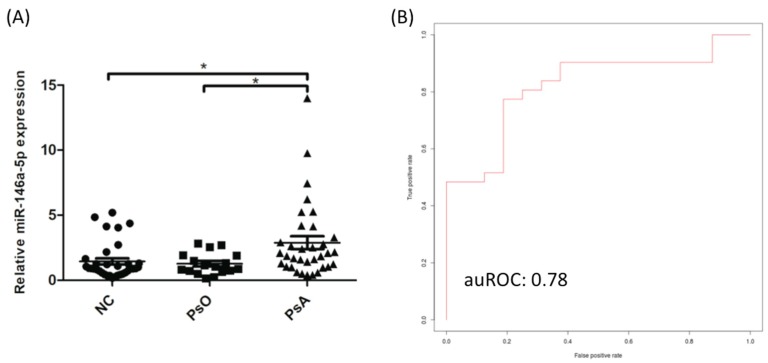
Increased expression of miR-146a-5p in CD14^+^ monocytes from PsA patients, validation by qRT-PCR (**A**), and demonstration of utility by support vector machine analysis (**B**). (**A**) The expression levels of miR-146a-5p were measured in NCs (*n* = 34), PsO patients (*n* = 17) and PsA patients (*n* = 34) by qRT-PCR. Patients with PsA showed increased expression of miR-146a-5p in CD14^+^ monocytes. (**B**) The Support Vector Machine (SVM) learning algorithm was used to distinguish PsA from NC. To diagnose PsA, a training model was generated by measuring miR-146a-5p expression in CD14^+^ monocytes from patients with PsA and NCs. The auROC was 0.78, indicating that miR-146a-5p expression can distinguish PsA patients from NCs with high accuracy. * *p* < 0.05.

**Figure 3 jcm-08-00110-f003:**
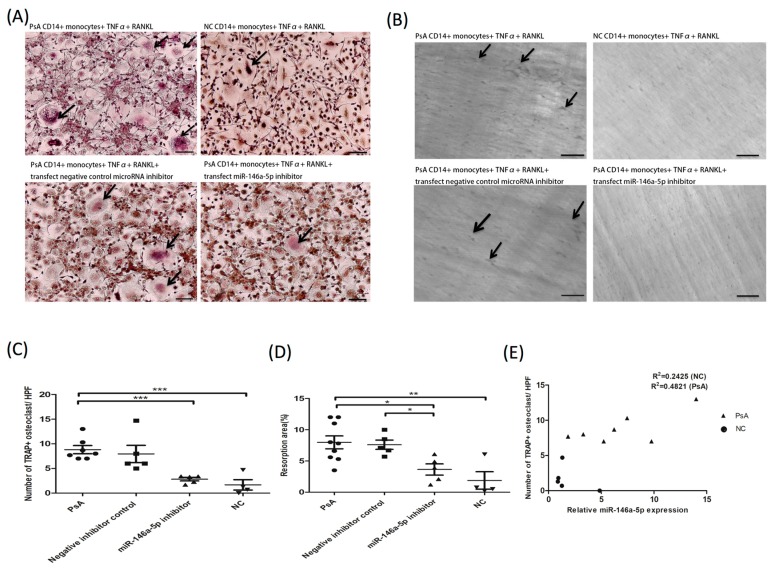
Preferential osteoclast differentiation and bone resorption activity in PsA patients were abrogated by miR-146a-5p inhibition. CD14^+^ monocytes from PsA patients (*n* = 10) and NCs (*n* = 5) were cultured with M-CSF 20 ng/mL for three days. PsA patient cells were then divided into three treatment groups: transfected with a control microRNA inhibitor (negative control), transfected with a miR-146 inhibitor, and non-transfected control. After transfection, cultures were treated with TNF-α and RANKL every three days, for nine days to induce osteoclast formation. On day 13, the number of osteoclasts formed and resorption pit area were measured. (**A**) Osteoclasts were identified by TRAP staining. Scale bar: 50 µm. (**B**) For evaluation of resorption activity, resorption pits on dentine slices were recorded under a bright field microscope from the average of four high power fields. Scale bar: 50 µm. (**C**) Osteoclasts was quantified from the average of four high power fields per group. (**D**) The eroded surface area on the dentine slice was quantified using ImageJ software and expressed as the % of total area. (**E**) Correlation of miR-146a-5p expression and osteoclast numbers was analyzed in PsA and NC group. * *p* < 0.05; ** *p* < 0.001; and *** *p* < 0.0001.

**Figure 4 jcm-08-00110-f004:**
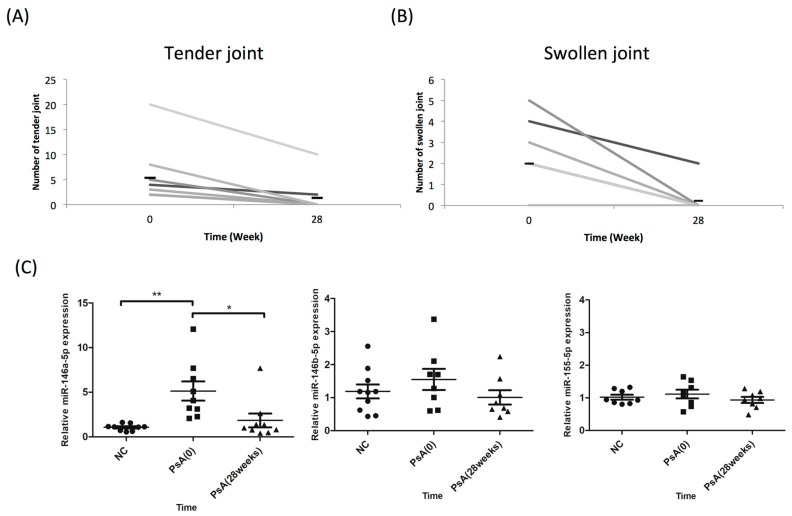
CD14^+^ monocytes from PsA patients showed reduced miR-146a-5p expression, but not miR-146b-5p or miR-155, following successful treatment. The expression levels of miR-146a-5p were measured in CD14^+^ monocytes derived from NCs and PsA patients both before and after successful treatment. (**A**) The number of tender joints before and after biological treatment for 28 weeks in patients with PsA. (**B**) The number of swollen joints before and after biological treatment for 28 weeks in patients with PsA. (**C**) miR-146a-5p, miR-146b-5p, and miR-155-5p expression levels were measured using qRT-PCR in CD14^+^ monocytes from 10 NCs and 10 patients with PsA, before and after treatment with biologics for 28 weeks. * *p* < 0.05; and ** *p* < 0.001.

**Figure 5 jcm-08-00110-f005:**
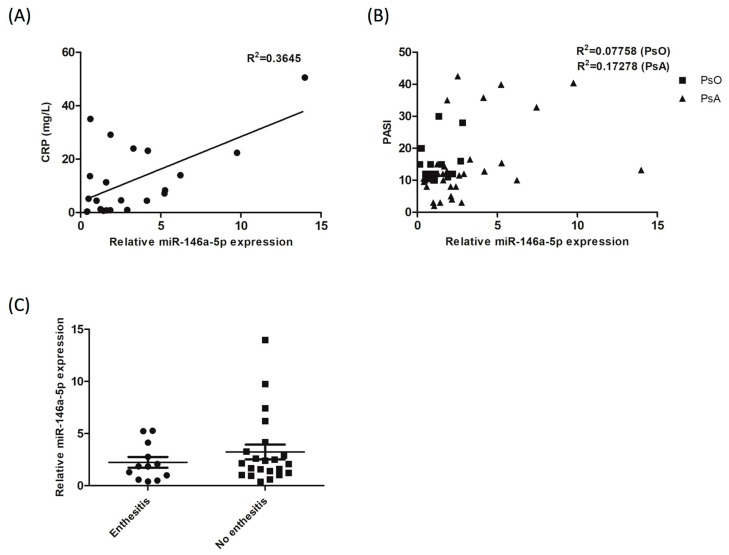
miR-146a-5p expression in CD14^+^ monocytes from PsA patients correlated to blood CRP level, but not the PASI score nor the presence of enthesitis. miR-146a-5p expression levels were measured in CD14^+^ monocytes from PsA and PsO patients. (**A**) CRP levels were measured in patients with PsA (*n* = 22). miR-146a-5p expression correlation with CRP levels was analyzed using the Pearson correlation coefficient. (**B**) The PASI severity score was measured in 34 PsA patients, and 17 PsO patients. The correlation of miR-146a-5p expression with PASI was analyzed using the Pearson correlation coefficient. (**C**) The presence of enthesitis was evaluated in 34 PsA patients. The expression of miR-146a-5p in patients with or without enthesitis was evaluated using a Student’s *t*-test.

**Table 1 jcm-08-00110-t001:** Demographics and clinical characteristics of psoriatic arthritis (PsA) patients, psoriatic patients without arthritis and normal controls.

	Patients with PsA (*N* = 34)	Psoriatic Patients without Arthritis (*N* = 17)	Normal Control (*N* = 34)
Age (years)	47.8 ± 11.2	43.2 ± 11.8	44.3 ± 12.1
Female—no. (%)	9 (26.5)	5 (29.4)	15 (44.0)
Weight—kg	73.4 ± 15.1	70.1 ± 11.5	66.0 ± 11.2
Psoriasis (years)	15.0 ± 8.3	15.1 ± 6.0	
Psoriatic arthritis (years)	8.1 ± 6.5		
No. of previous anti-TNFα drugs—no. (%)	8 (23.5)	8 (23.5)	
Use of methotrexate—no. (%)	28 (82.4)	13 (76.5)	
Use of leflunomide	10 (29.4)		
Use of NSAID	30 (88.2)		
Patients with specific disease characteristics—no. (%) CRP (mg/L)	11.9 ± 13.2		
PASI	14.8 ± 11.3		
Peripheral arthritis	21 (61.8)		
Peripheral and axil arthritis	13 (38.2)		
Dactylitis	8 (25.8)		
Enthesitis	12 (41.9)		
Tender-joint count (of 68 joints)	9.1 ± 9.3		
Swollen-joint count (of 66 joints)	3.5 ± 6.8		
Uveitis	2 (5.9)		

PsA: psoriatic arthritis; NSAID: nonsteroidal anti-inflammatory drug; CRP: C-Reactive Protein; PASI: Psoriasis Area and Severity Index.

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
