# Peer review of "MiR-146a-5p Expression in Peripheral CD14+ Monocytes from Patients with Psoriatic Arthritis Induces Osteoclast Activation, Bone Resorption, and Correlates with Clinical Response"

_jcm, 2019, doi:10.3390/jcm8010110_

Reviewer 1 Report

This is an interesting study whereby authors concluded that miR-146a-5p can correlate with disease severity in PsA and thus can serve as early diagnostic marker. There are many major and minor concerns that are outlined below.

Major concerns

The data is not adequately supporting authors’ conclusions. miR-146a-5p can be simply correlating with inflammation in patients and might not be necessarily representative of disease severity only in PsA patients. Authors need to include patients with psoriasis (without arthritis) show the miR-146a-5p is specific for PsA.

Based on the presented data authors cannot claim miR-146a-5p can be a diagnostic marker. To do that they need to have samples from other arthritis such as rheumatoid arthritis, spondyloarthritis, osteoarthritis (at least 2-3 types).

Authors showed that biologic treatment reduced miR146a expression levels in PsA patients. Authors also need to present what were the expression patterns for the other miRNAs.

It is not clear how the figure 1A and 2A are different and why the differences for same miRNA present in these two figures (figure 1 and Figure 2).

The authors did not explain the rationale behind this work and how they decided to study only miR-146a-5p, miR-146b-5p, and miR-155 and no other miRNAs.

The authors did not establish well what is the correlation between miR-146a-5p and osteoclast precursors (OCP) numbers. It is already known that PsA patients have higher amounts of OCP, so it comes as no surprise. Moreover, authors did not clearly show how miR-146a-5p and OC number significantly different between HC and PsA patients. They even did not present the standard deviation or standard error clearly in Figure 3D.

Authors also need to present the correlation between miR-146a-5p and any inflammation marker such as CRP.

Minor concerns

It is not clear why the authors used TNF with RANKL for OC and how these experiments are helping their conclusion (data do not add little knowledge due to the limited experiment was done).

Authors need to present the data from each patient with individual measures for their data presented as bar graphs.

What is delta delta Cq (or authors meant delta delta Ct)?

Overall comments

The presented work is interesting but not the presented data are not adequately convincing and do not support the author's' major conclusions. Need more experimentations and better controls.

Author Response

Dear reviewer:

Thanks for your efforts and comments. We have revised the manuscript accordingly. Please see our point to point responses below (also see in attached file). 

Comments of reviewer 1:

Major concerns:

1. The reviewer’s concern that the data is not adequately supporting authors’ conclusions. miR-146a-5p can be simply correlating with inflammation in patients and might not be necessarily representative of disease severity only in PsA patients. Authors need to include patients with psoriasis (without arthritis) show the miR-146a-5p is specific for PsA.

Response:

Thanks for the comments.

We have incorporated additional 17 patients with psoriasis without arthritis (PsO), along with the original 34 patients of psoriatic arthritis (PsA) and 34 healthy controls (NC), in the study. The data consistently showed that miR-146-5p expression in CD14+ cells from patients with PsA was significantly higher than that from patients with either PsO or NC. Accordingly, title of this manuscript is revised as ‘’MiR-146a-5p expression in peripheral CD14+ monocytes from patients with psoriatic arthritis induces osteoclast activation, bone resorption, and correlates with clinical efficacy’’. We have incorporated the demographic data of psoriatic patient without arthritis to Table 1 and we also have incorporated the data of miR-146a-5p expression in patients with PsO into Figure 2. Please refer to the revised Table 1(P14~15) and Figure 2(P17).

2. Based on the presented data authors cannot claim miR-146a-5p can be a diagnostic marker. To do that they need to have samples from other arthritis such as rheumatoid arthritis, spondyloarthritis, osteoarthritis (at least 2-3 types).

Response:

We repeated the experiments in a small scale from 3 patients with rheumatoid arthritis (RA) and 3 patients with ankylosing spondylitis (AS).

The results showed that the expression of miR-146a-5p in patients with RA and AS was similar to that in the NC group and significantly lower than that in PsA (as the figure in attached file). Since the numbers of RA and AS were small and osteoarthritis (OA) was not taken into consideration, we did not incorporate these two groups in the study and we replaced the title of this talk as ’MiR-146a-5p expression in peripheral CD14+ monocytes from patients with psoriatic arthritis induces osteoclast activation, bone resorption, and correlates with clinical efficacy”.

3. Authors showed that biologic treatment reduced miR146a expression levels in PsA patients. Authors also need to present what were the expression patterns for the other miRNAs.

Response:

In addition to the miR-146a-5p, we also analyzed the expressions of miR-146b and miR-155-5p from CD14+ monocytes in NC, patients with PsA before and after 28 weeks biologic treatment(n=31, 10 and 10). The result showed that only miR-146a-5p, but not miR-146b-5p nor miR-155, was reduced in parallel to the clinical improvement. Please refer to Figure 4C.

4. It is not clear how the figure 1A and 2A are different and why the differences for same miRNA present in these two figures (figure 1 and Figure 2).

Response:

The y-axis is different in Figure 1A and 2A, one is relative expression level and one is the delta Ct representing cycle numbers in PCR. The y-axis as expression level has been unified in all the figures. Please refer to the revised Figure 1 and 2 (P16~18).

5. The authors did not explain the rationale behind this work and how they decided to study only miR-146a-5p, miR-146b-5p, and miR-155 and no other miRNAs.

Response:

More than 40 miRNAs (including miR-21, -29, -31, -124, -133a, -146a, -223, -503, -378, -125a, -148a, -155, and -422a) have been documented to regulate the differentiation of osteoclast precursors into mature osteoclasts. On the other hand, many of the miRNAs also regulated the osteoblast differentiation with known transcription factors and epigenetic regulators, including miR-155, -30, -503, -146a, and -541.

Experimental evidences showed that Mir-146a inhibits proliferation and induces apoptosis through bcl-2 in osteoblasts. A study in PNAS also showed that miR-155 is upregulated in CD68+ macrophages in the synovium from RA. Hence, we decided to choose the common miRNAs, miR-155 and miR-146a, in the osteoclastic and osteoblastic differentiation to start with since PsA is characterized by both osteoclastic and osteoblastic activation. The other similar form of miR-146a, the miR-146b, was also chosen as an internal control. Please refer to the introduction section.

In fact, more than 40 miRNAs (including miR-21, -29, -31, -124, -133a, -146a, -223, -503, -378, -125a, -148a, -155, and -422a) have been documented to regulate the differentiation of osteoclast

precursors into mature osteoclasts (23). On the other hand, many of the miRNAs also regulated the osteoblast differentiation with known transcription factors and epigenetic regulators, including miR-155, -30, -503, -146a, and -541(24). Experimental evidences showed that Mir-146a inhibits proliferation and induces apoptosis through bcl-2 in osteoblasts(25). A study published in PNAS also showed that

miR-155 is upregulated in CD68+ macrophages in the synovium from RA(26). Hence, we decided to choose the two common miRNAs, miR-155 and miR-146a, in the osteoclastic and osteoblastic differentiation to start with since PsA is characterized by both osteoclastic and osteoblastic activation. The other similar form of miR-146a, the miR-146b, was also chosen as an internal control.

6. The authors did not establish well what is the correlation between miR-146a-5p and osteoclast precursors (OCP) numbers. It is already known that PsA patients have higher amounts of OCP, so it comes as no surprise. Moreover, authors did not clearly show how miR-146a-5p and OC number significantly different between HC and PsA patients. They even did not present the standard deviation or standard error clearly in Figure 3D.

Response:

We asked whether the number of osteoclast differentiation would be associated with the expression of miR-146a-5p in patients with PsA. The R Square of PsA and NC group are 0.4821 (p=0.08) and 0.2425(p=0.41), respectively, indicating a small but significant association between miR-146a-5p expression and the number of osteoclasts in PsA. Please refer to Figure 3E (or in attached file).

7. Authors also need to present the correlation between miR-146a-5p and any inflammation marker such as CRP.

Response:

We asked whether the expression of miR-146a in CD14+ monocytes was associated with the blood CRP level in 22 patients with PsA. The result showed that R Square is 0.3645(p=0.0029)(Figure 5A),

indicating a small but significant association of miR-146a-5p in CD14+ monocytes with blood CRP level in patients with PsA. Please refer to Figure 5A (or in attached file).

Minor concerns:

1. It is not clear why the authors used TNF with RANKL for OC and how these experiments are helping their conclusion (data do not add little knowledge due to the limited experiment was done).

Response:

In 1998, two different research groups reported that reception activation of NF-kB ligand (RANKL) was essential for osteoclast differentiation (Proceedings of the National Academy of Sciences of the United States of America, vol. 95, no. 7, pp. 3597–3602, 1998; Cell, vol. 93, no. 2, pp. 165–176, 1998). RANKL induces osteoclast differentiation by binding to RANK in myeloid cells and monocyts. TNF-a was reported to induce the formation of osteoclasts from bone marrow macrophages in vitro. TNF-a was shown to permit the osteoclast activation by RANKL as reported in J Clin Invest in 2000 (Journal of Clinical Investigation, vol. 106, no. 12, pp. 1481-1488, 2000). Hence, to induce osteoclast differentiation from circulating monocytes in this study, RANKL and TNF-a were used to induce osteoclastogenesis as a model.

2. Authors need to present the data from each patient with individual measures for their data presented as bar graphs.

Response:

Thanks for the comments. The bar graph has been replaced by the dotted plots. Please refer to the revised Figures.

3. What is delta delta Cq (or authors meant delta delta Ct)?

Response:

Actually, the delta Cq was a typo of delta Ct. However, for standardization and consistence, the relative expression levels are used throughout the figures to avoid confusion between delta Ct and expression levels.

Overall comments

The presented work is interesting but not the presented data are not adequately convincing and do not support the author's' major conclusions. Need more experimentations and better controls.

Response:

As pointed out earlier in the reply, the title has been replaced to avoid the use of diagnostic biomarker. In addition, patients with psoriasis without arthritis (n=17), two more small groups from RA and AS (n=3 and 3, respectively) were used to compare their expression levels of miR-146a-5p in PsA and in NCs. There was an association of miR-146a-5p expression and CRP level from patients with PsA.

We hope that the revised manuscript in its current form is of general interest for the readers and is appropriate for publication in Journal of Clinical Medicine.

With best regards,

Chih-Hung Lee, MD, PhD.

Department of Dermatology, Kaohsiung Chang Gung Memorial

Hospital, Kaohsiung, Taiwan

Address: Ta-Pei Road 123, Niao-Sung district, Kaohsiung 83301, Taiwan

Telephone: +886-7-7317123 ext 2299

Reviewer 2 Report

 An interesting study addressing the role of miR-146a-5p in the PsA pathogenesis by comparing the levels of mir-146a-5p in peripheral CD14+ 2 monocytes in a small group of 34 patients with active PsA patients compare to controls and in comparison to the clinical response in the PsA group as well.

In this aspect the study design and results meet its aims but it does not prove the role of miR 146 as a biomarker. The study population did not included other inflammatory arthritis as rheumatoid arthritis in one hand or psoriasis without arthritis in the other hand. Therefore the article title, discussion and conclusion should be revised.

The correlation to disease activity is limited to the improvement in  joint disease only as measured by ACR 20 and not to the other manifestations of the disease as PASI ,  enthesitis etc. If data is available it will interesting to assess the changes in mir 146 in correlation to those parameters or to the minimal disease activity (MDA )score as well.

Author Response

Dear reviewer:

Thanks for your efforts and comments. We have revised the manuscript accordingly. Please see our point to point responses below (also in attached file). 

Reviewer 2

Comments and Suggestions for Authors

1. An interesting study addressing the role of miR-146a-5p in the PsA pathogenesis by comparing the levels of mir-146a-5p in peripheral CD14+ 2 monocytes in a small group of 34 patients with active PsA patients compare to controls and in comparison to the clinical response in the PsA group as well.

In this aspect the study design and results meet its aims but it does not prove the role of miR-146 as a biomarker. The study population did not include other inflammatory arthritis as rheumatoid arthritis in one hand or psoriasis without arthritis in the other hand. Therefore the article title, discussion and conclusion should be revised.

Response:

Thanks for the comments. The title has been replaced to avoid the use of diagnostic biomarker. In addition, patients with psoriasis without arthritis (n=17), two more small groups from RA and AS (n=3 and 3, respectively) were used to compare their expression levels of miR-146a-5p in PsA and in NCs. The title has been replaced as ‘’MiR-146a-5p expression in peripheral CD14+ monocytes from patients with psoriatic arthritis induces osteoclast activation, bone resorption, and correlates with clinical efficacy’’. The sections of discussion and conclusion are also revised.

2. The correlation to disease activity is limited to the improvement in joint disease only as measured by ACR 20 and not to the other manifestations of the disease as PASI, enthesitis etc. If data is available it will interesting to assess the changes in mir 146 in correlation to those parameters or to the minimal disease activity (MDA) score as well.

Response:

Thanks for the useful comments. Unfortunately, we do not have MDA score available, however, we do have PASI and presence of enthesitis.

The association was analyzed in 34 patients with PsA and 17 patients with PsO. The result showed R Square in PsA and PsO patients is 0.17(p=0.08) and 0.07(p=0.50), respectively (Figure 5B), indicating a non-association of miR-146a-5p expression with PASI severity score in patients with PsA or PsO.

Finally, we ask whether PsA patients with enthesitis would have an increased expression of miR-146a-5p expression. The expression of miR-146a-5p was measured in 12 PsA patients with enthesitis and 22 PsA patients without enthesitis. The result showed the expression of miR-146a-5p in patients with or without enthsitis was similar (p=0.35) (Figure 5C). Please refer to Figure 5B and 5C (also in attached file).

We hope that the revised manuscript in its current form is of general interest for the readers and is appropriate for publication in Journal of Clinical Medicine.

With best regards,

Chih-Hung Lee, MD, PhD.

Department of Dermatology, Kaohsiung Chang Gung Memorial

Hospital, Kaohsiung, Taiwan

Address: Ta-Pei Road 123, Niao-Sung district, Kaohsiung 83301, Taiwan

Telephone: +886-7-7317123 ext 2299

Reviewer 3 Report

This small study reports on the expression of a cell marker in patients with established psoriatic arthritis compared to healthy controls. The authors claim that this could serve as an early biomarker for the disease. Unfortunately, they can’t make this claim from the data they present. We would need to see other cohorts – people with psoriasis without musculoskeletal symptoms, early untreated psoriatic arthritis, different subgroups of psoriatic arthritis, and patients with other inflammatory diseases especially rheumatoid arthritis. The best they can claim is that the biomarker is present in established disease (v HC) and it diminishes after successful treatment. Please also note that CASPAR criteria are for classification, not diagnosis.

Author Response

Dear reviewer: 

Thanks for your efforts and comments. We have revised the manuscript accordingly. Please see our point to point responses below (also in attached file).

Comments and Suggestions for Authors

This small study reports on the expression of a cell marker in patients with established psoriatic arthritis compared to healthy controls. The authors claim that this could serve as an early biomarker for the disease. Unfortunately, they can’t make this claim from the data they present. We would need to see other cohorts – people with psoriasis without musculoskeletal symptoms, early untreated psoriatic arthritis, different subgroups of psoriatic arthritis, and patients with other inflammatory diseases especially rheumatoid arthritis. The best they can claim is that the biomarker is present in established disease (v HC) and it diminishes after successful treatment. Please also note that CASPAR criteria are for classification, not diagnosis.

Response:

Thanks for the useful comments. The title has been replaced to avoid the use of diagnostic biomarker. In addition, patients with psoriasis without arthritis (n=17), two more small groups from RA and AS (n=3 and 3, respectively) were used to compare their expression levels of miR-146a-5p in PsA and in NCs. The title has been replaced as ‘’MiR-146a-5p expression in peripheral CD14+ monocytes from patients with psoriatic arthritis induces osteoclast activation, bone resorption, and correlates with clinical efficacy". We also revised the manuscript in the section of discussion and conclusion to avoid the use of disease biomarkers for miR-146a-5p.

We hope that the revised manuscript in its current form is of general interest for the readers and is appropriate for publication in Journal of Clinical Medicine.

With best regards,

Chih-Hung Lee, MD, PhD.

Department of Dermatology, Kaohsiung Chang Gung Memorial Hospital, Kaohsiung, Taiwan

Address: Ta-Pei Road 123, Niao-Sung district, Kaohsiung 83301, Taiwan

Telephone: +886-7-7317123 ext 2299

Round  2

Reviewer 1 Report

Manuscript is much better now.

Author Response

Dear reviewer:

Thanks for your efforts and comments. The manuscript has been edited by the MDPI English editing service.

We hope that the revised manuscript in its current form is of general interest for the readers and is appropriate for publication in Journal of Clinical Medicine.

Best regards,

Chih-Hung Lee, MD, PhD.

Department of Dermatology, Kaohsiung Chang Gung Memorial Hospital, Kaohsiung, Taiwan

Address: Ta-Pei Road 123, Niao-Sung district, Kaohsiung 83301, Taiwan

Telephone: +886-7-7317123 ext 2299

Reviewer 2 Report

All my comments were addressed.

Author Response

(The authors gave the same response as above.)

Reviewer 3 Report

 MiR-146a-5p Expression in Peripheral CD14+  Monocytes from Patients with Psoriatic Arthritis Induces Osteoclast Activation, Bone Resorption, and Correlates with Clinical Efficacy

This paper has been improved but still has a lot of limitations. I suggest they change the title to:

 MiR-146a-5p Expression in Peripheral CD14+ Monocytes from Patients with Psoriatic Arthritis Induces Osteoclast Activation, Bone Resorption, and Correlates with Clinical Response

I don’t see data from RA and AS in here

The numbers are small. Look at figure 2. Most of the PsA patients do not have an elevated marker. What distinguishes those who do? This is very preliminary data and should be treated as such.

There are still areas where they use the word ‘predictor’ eg the heading on page 182

Author Response

Dear reviewer:

Thanks for your efforts and comments. We have revised the manuscript accordingly. Please see our point to point responses below. 

Comments of reviewer:

1. The reviewer suggests moderate English changes in this manuscript.

Response:

Thanks for the comments. The manuscript has been edited by the MDPI English editing service.

2.This paper has been improved but still has a lot of limitations. I suggest they change the title to: MiR-146a-5p Expression in Peripheral CD14+ Monocytes from Patients with Psoriatic Arthritis Induces Osteoclast Activation, Bone Resorption, and Correlates with Clinical Response

Response:

Per the reviewer’s suggestion, we change the title of this manuscript to ‘’MiR-146a-5p Expression in Peripheral CD14+ Monocytes from Patients with Psoriatic Arthritis Induces Osteoclast Activation, Bone Resorption, and Correlates with Clinical Response‘’ Please refer to Line 5.

3. The reviewer does not see data from RA and AS.

Response:

During the one-week time limit for the last revision, we only had a chance to repeat the experiment in a small scale from 3 patients with rheumatoid arthritis (RA) and 3 patients with ankylosing spondylitis (AS). The results showed that the expression of miR-146a-5p in patients with RA and AS was similar to that in the NC group but significantly lower than that in PsA (as the figure in attachment). Since the numbers of RA and AS were small and the patients with osteoarthritis (OA) were not taken into consideration, we did not incorporate the data from these two groups (RA and AS) in the manuscript. Please refer to the Figure below, showing the selective increase of miR-146a-5p level in patients with PsA.

4. The numbers are small. Look at figure 2. Most of the PsA patients do not have an elevated marker. What distinguishes those who do? This is very preliminary data and should be treated as such.

Response:

In Figure 2, it is true that some, but not most, of the patients with PsA do not have the elevated marker. In fact, using the average value of miR-146a-5p in normal controls as the cut point, the expression of miR-146a-5p is increased in 80% of PsA patients (28/34) but only in 23% (8/34) of the normal controls. Although the subject numbers are not big (n=34 each), based on the statistical comparisons between PsA and normal controls, and computer-assisted support vector machine analysis, there are significant differences in miR-146a-5p expressions between the two groups. For sure the expression of miR-146a-5p is not a perfect 100% diagnostic marker to distinguish PsA from normal controls, however, its accuracy to distinguish PsA and healthy controls (auROC level) is 0.78, indicating that miR-146a-5p expression is able to distinguish PsA patients from NCs in 78% of the cases. In parallel, our result also showed that miR-146a-5p expression was reduced in CD14+ cells from PsA patients during clinical remission, indicating its expression could reflect the disease severity in PsA. Please refer to Line 198-202.

To investigate if miR-146a-5p levels could distinguish PsA patients from NCs, we used support vector machine (SVM) learning [29,30] to calculate the discrimination power of miR-146a-5p. We generated a SVM model with an area under the receiver operating curve (auROC) of 0.78 (Figure 2B), indicating that miR-146a-5p expression could distinguish PsA patients from NCs in 78% of the cases.

5. There are still areas where they use the word ‘predictor’ eg the heading on page 182

Response:

We discarded the word ‘predictor’ in Line 182 according to your comment. In addition, the heading of "qPCR Coupled with Machine Learning Identified miR-146a-5p as An Early Predictor for PsA" is replaced to the heading of "The Expression of miR-146a-5p was Significantly Increased in CD14+ Monocytes from PsA Patients compared to PsO Patients and NCs’’. Please refer to Line 190~191.

The Expression of miR--‐146a--‐5p was Significantly Increased in CD14+ Monocytes from PsA Patients compared to PsO Patients and NCs.

In the discussion section (Line 303~304), the sentence of "Our study identifies miRNA-146a-5p in CD14+ monocytes as an easily accessible marker for prompt computer-aided diagnosis of PsA (with an auROC value 0.78 by SVM model)" using the term "marker" has been deleted.

We hope that the revised manuscript in its current form is of general interest for the readers and is appropriate for publication in Journal of Clinical Medicine.

Best regards,

Chih-Hung Lee, MD, PhD.

Department of Dermatology, Kaohsiung Chang Gung Memorial Hospital, Kaohsiung, Taiwan

Address: Ta-Pei Road 123, Niao-Sung district, Kaohsiung 83301, Taiwan

Telephone: +886-7-7317123 ext 2299
